# Glyphosate Efficacy in *Chloris virgata* Sw. in Response to Temperature and Tank Mixing

**DOI:** 10.3390/plants11233190

**Published:** 2022-11-22

**Authors:** Gulshan Mahajan, Bhagirath Singh Chauhan

**Affiliations:** 1The Centre for Crop Science, Queensland Alliance for Agriculture and Food Innovation (QAAFI), The University of Queensland, Gatton, QLD 4343, Australia; 2Department of Agronomy, Punjab Agricultural University, Ludhiana 141004, Punjab, India; 3School of Agriculture and Food Sciences (SAFS), The University of Queensland, Gatton, QLD 4343, Australia; 4Department of Agronomy, Chaudhary Charan Singh Haryana Agricultural University (CCSHAU), Hisar 125004, Haryana, India

**Keywords:** dose–response curve, herbicide efficacy, herbicide resistant population, high temperature, low temperature, weed control, 2,4-D

## Abstract

Glyphosate alone or a tank mixture of glyphosate and 2,4-D is commonly used for broad-spectrum weed control under fallow conditions in Australia. Air temperature or mixing glyphosate with 2,4-D, may influence the efficacy of glyphosate on feather fingergrass (*Chloris virgata* Sw.), a problematic summer-season weed of Australia. Dose–response studies were conducted with four populations of feather fingergrass under temperature-controlled glasshouse conditions (35/25 °C and 25/15 °C at 12 h/12 h) to assess the level of glyphosate resistance in relation to temperature regimes. Four parameter log-logistic models were used to develop dose–response curves. Based on plant mortality percentage, LD_50_ (lethal dose for 50% mortality) values of glyphosate at 25/15 °C for populations Ch, SGM2, SGW2, and CP2 were 137, 60, 650, and 1067 g ae ha^−1^, respectively. However, at 35/25 °C, the corresponding LD_50_ values were 209, 557, 2108, and 2554 g ae ha^−1^, respectively. A similar response was observed for the parameter GR_50_ (dose for 50% growth reduction) values of glyphosate. These results indicate that populations SGW2 and CP2 are highly glyphosate-resistant and in the summer season, it may be very difficult to control these populations due to poor glyphosate efficacy. These results further suggest that the efficacy of glyphosate for feather fingergrass control can be improved if applied during cooler temperatures (25/15 °C) or the spring season compared with warmer temperatures (35/25 °C) or the summer season. In another study, 2,4-D antagonized glyphosate remarkably in the CP2 (glyphosate-resistant) population but only marginally in the Ch (glyphosate-susceptible) population. Thus, it is not advisable to mix 2,4-D with glyphosate for the control of glyphosate-resistant feather fingergrass populations. The results further suggest that the use of this mixture is useful if the feather fingergrass is not glyphosate-resistant; however, the use of the mixture is to be avoided if the population is glyphosate-resistant in order to not exacerbate the potential resistance problem.

## 1. Introduction

Feather fingergrass (*Chloris virgata* Sw.) is a problematic summer-season weed in southeastern Australia. This weed is well adapted to the no-till production system in Australia and has evolved resistance to glyphosate [1,2]. During the 1990s, the problem of *C. virgata* was limited to some pockets of Australia. However, with the increased adoption of the no-till production system, it spread throughout the country [3]. In Australia, the infestation of *C. Virgata* was estimated on an area of 0.2 mha [4]. Light-stimulated germination and high germination in the soil surface layers are the main ecological factors for the adaptation of *C. virgata* in the no-till system [5,6,7]. *Chloris virgata* seeds can germinate under a wide range of temperatures; therefore, this weed can emerge throughout the year in Australian paddocks [6,8]. The seed production potential of *C. virgata* is very high and one plant can produce >140,000 seeds under fallow conditions [9]. The dispersal ability of *C. virgata* is also high as its seeds are light in weight and have trichomes [6]. These attributes have enabled the wide distribution of this weed throughout Australia. As a result of these characteristics, *C. virgata* is now included in the category of the top 20 weeds of major concern to Australia [4]. In a recent study, an infestation of 22–25 plants m^−2^ of *C. virgata* caused a 50% grain yield reduction in mungbean compared with a weed-free environment [10].

A fallow phase between two crops is a common practice in Australia. Farmers, in general, use glyphosate for weed control during the fallow phase and avoid tillage to save the residual soil moisture and nutrients for subsequent crops. The frequent use of glyphosate under fallow conditions for weed control has led to the problem of glyphosate resistance in *C. virgata* [1,11]). The first glyphosate-resistant (GR) case of *C. virgata* in Australia was reported in 2013 and afterward, many GR populations have been reported [1,2,7]. Recently, it was found that *C. virgata* populations collected from Queensland have evolved target-site resistance to glyphosate [12]. The resistance level of these populations to glyphosate could be due to their varied emergence time and prevailing temperature conditions at the time of glyphosate application. It has been reported that glyphosate resistance levels in some weeds (junglerice, *Echinochloa colona* (L.) Link; hairy fleabane, *Conyza bonariensis* (L.) Cronq.; and annual sowthistle, *Sonchus oleraceus* L.) varied with temperature, and glyphosate efficacy against these weeds was reduced at high temperatures [13,14,15,16]. Contrary to this, it has been reported that the efficacy of glyphosate for controlling common (*Ambrosia artemisiifolia* L.) and giant ragweed (*Ambrosia trifida* L.) improved if applied during warm temperatures (29/17 °C d/n) compared with cooler temperatures (20/11 °C d/n) due to increased absorption and/or translocation [17].

For weed control in fallow situations, herbicide tank mixing is a common practice in Australia. For example, tank mixing of glyphosate and 2,4-D is a commonly used practice for the control of grass and broadleaf weeds, particularly when the weed mixture is composed of GR broadleaf weeds, such as hairy fleabane and sowthistle. In such situations, the herbicide mixture can provide broad-spectrum (grass as well as broad-leaf weeds) weed control, reduce the cost of weed control, and delay the evolution of herbicide resistance [18]. However, synergistic or antagonistic (the synergistic effect is when herbicides are applied in a mixture and weed control is less than expected compared with when herbicides are applied alone; however, the inverse response is described as being antagonistic) effects of herbicides on weeds may occur when two or more herbicides are used as a mixture [19,20]. This synergistic or antagonistic effect might be due to altered herbicide absorption and translocation [21,22], suggesting that the effect of herbicides on weeds could be weed-specific. For bindweed (*Convolvulus arvensis* L.) control, a synergistic effect of 2,4-D and glyphosate has been reported due to greater glyphosate translocation to roots [19]. Contrary to this, an antagonistic effect of 2,4-D and glyphosate in Johnsongrass (*Sorghum halepense* (L.) Pers.) was reported due to reduced absorption and translocation [23]. Similarly, 2,4-D (amine/ester) antagonized the effect of glyphosate in controlling junglerice [24]. In another study, the auxin herbicide dicamba antagonized the effect of glyphosate in controlling *Kochia scoparia* due to poor translocation [22].

Information is not available to understand whether a mixture of glyphosate and 2,4-D has antagonistic or synergistic effects on *C. virgata* control. Information is also lacking on whether temperature plays a role in variable resistance levels to glyphosate in different populations of *C. virgata*. Therefore, a study was conducted in an automatic temperature-controlled glasshouse to compare glyphosate resistance levels in different populations of *C. virgata* at different temperature regimes (35/25 and 25/15 °C). A second study was conducted in an open space to assess the synergistic or antagonistic effects of 2,4-D and glyphosate for the control of *C. virgata* populations.

## 2. Materials and Methods

### 2.1. Seed Collection

Seeds of four populations of *C. virgata* were collected from different paddocks (randomly without knowing the resistance status) of south-eastern Australia in March–April, 2017 (Table 1). For each population, seeds were selected from 50–60 random plants that were spread about 500 × 500 m^2^ in each paddock. After sun drying, seeds were stored under dark conditions at room temperature (25 ± 2 °C) in the weed science laboratory of the Queensland Alliance for Agriculture and Food Innovation (QAAFI), The University of Queensland, Australia. For experimental purposes, seeds of each population were used to grow plants in pots under an open field environment at the Gatton Research Farm in January 2019 and fresh seeds of each population (random plants of same population) were collected. Seeds were stored under dark conditions at room temperature in the laboratory until being used for experimental purposes.

### 2.2. Experiment 1. Effect of Temperature on Glyphosate Efficacy

To evaluate the effect of temperature on glyphosate efficacy for *C. virgata* control, a pot study was conducted in two automatic temperature-controlled glasshouse bays. One glasshouse bay was maintained at a high temperature (HT) regime, i.e., day/night temperature of 35/25 °C (12 h/12 h), and the second glasshouse bay was maintained at a low temperature (LT) regime, i.e., day /night temperature of 25/15 °C (12 h/12 h). The humidity in both bays was kept constant and ranged between 60 and 70%. Experimental treatments included (i) first factor: two temperature regimes (35/25 °C and 25/15 °C), (ii) second factor: four populations (Ch, SGM2, SGW2, and CP2) of *C. virgata*, and (iii) third factor: eight glyphosate doses (0, 143, 285, 570, 1140, 2280, and 4560 g ae ha^−1^). These treatments were tested in a factorial arrangement of a randomized complete block design in three replicates. The first experimental run was started in January 2021, and the second experimental run was started in March 2021.

The study was conducted in pots (20 cm diameter) filled with potting mix (Centenary Landscape, Australia) and in each pot, four plants were grown. Initially, 20 seeds per pot were planted and after emergence, only four plants per pot were kept. Glyphosate doses were applied with the help of a research track sprayer (manufactured by Woodlands Road Engineering, Gatton, Australia), delivering a spray volume of 108 L ha^−1^ through flat fan nozzles (TeeJet XR 110015). The spray application was performed when the plants attained the 6–8 leaf stage. Both experimental runs were terminated 28 d after spray.

### 2.3. Experiment 2. Antagonistic Effect of 2,4-D on Glyphosate Efficacy

In this study, two populations of *C. virgata,* namely, Ch (glyphosate-susceptible (GS)) and CP2 (glyphosate-resistant (GR)), were selected. The GS population was tested at six glyphosate rates (0, 92.5, 185, 370, 740, and 1482 g ae ha^−1^), whereas the GR population was tested at seven different glyphosate rates (0, 370, 741, 1482, 2223, 2964, and 3705 g ae ha^−1^). The GR population was highly resistant to glyphosate; therefore, higher rates of glyphosate were used for this population.

The first experimental run of this study was started in October 2020 (with six glyphosate rates for Ch and seven glyphosate rates for CP2) and terminated in December 2020. The second experimental run was started in January 2021 and terminated in February 2021. All treatments were evaluated in a factorial arrangement of a randomized complete block design in three replicates by keeping four seedlings per pot. The first factor in the experimental design was population and the second factor was glyphosate rates. The study was conducted in pots (20 cm diameter) filled with potting mix (Centenary Landscape, Australia) and in each pot, four plants were grown. The pots were kept on benches and maintained in an outdoor environment. The mean monthly maximum and minimum temperature in the first experimental run ranged between 30 and 33 °C and 16 and 20 °C, respectively. The mean monthly maximum and minimum temperature in the second experimental run ranged between 32 and 33 °C and 18 and 19 °C, respectively. When plants attained the 10–12 leaf stage (~20 d after planting), they were treated with glyphosate alone or with a tank mixture of glyphosate and 2,4-D amine salt (700 g ae ha^−1^; at the label rate) as per treatments. No foam, gel, or precipitation was observed during the tank mixing. The spray was applied using the sprayer as mentioned above.

### 2.4. Observation Recorded

The mortality percentage for treated plants in each experiment was assessed based on a comparison with nontreated control plants of their respective populations. Plants were considered dead if no new growth or green tissue appeared after 28 d of herbicide application. For aboveground biomass measurements, plants in each pot were cut close to the base at 28 d after treatment, oven-dried at 70 °C for 72 h, and weighed per pot. The biomass data were converted into percent biomass reduction compared with the nontreated control of the respective population in each experiment.
Percent biomass reduction = [(A − B)/A] × 100 (1)
where A is the mean biomass of the nontreated control of each population in each temperature regime or herbicide mixture treatment, and B is the mean biomass of each population at the herbicide dose *x* in each temperature regime or herbicide mixture treatment. The resistance index (resistance/susceptible) ratio was compared on the basis of LD50/GR50 to compare the resistance level between the two populations. For calculating the resistance index, the population Ch was considered as the susceptible population.

### 2.5. Statistical Analyses

#### 2.5.1. Experiment 1. Effect of Temperature on Glyphosate Efficacy

Analysis of variance (ANOVA) was performed using Genstat (16th Edition; VSN International, Hemel Hempstead, UK). The trend was similar in both experimental runs and there was no interaction between treatments and experimental runs, therefore, the data were pooled over two experimental runs. Treatment means were separated at *p* ≤ 0.05 using Fisher’s protected least significant differences (LSD) test. Mortality and biomass reduction (as a percentage compared to the nontreated control) data were regressed over herbicide treatments using a four-parameter log-logistic model using SigmaPlot 14.0 (Systat Software, San Jose, CA, USA).
*y* = *y*_0_ + [*a*/1 + (*x*/*x*_50_)*^b^*](2)
where *y* = mortality (%) or biomass reduction (%); *y*_0_ = bottom of the curve; *a* = difference between the top and bottom of the curve; *x*_50_ = dose required to kill 50% plants (LD_50_; based on visual control) or plant growth (GR_50_; based on plant biomass); *b* = slope of the curve; and *x* = glyphosate dose. The fitness of the selected model was determined using *R*^2^ values (best fit). All raw data and tables along with LSD values in relation to Experiments 1 and 2 are provided in the Appendix A.

#### 2.5.2. Experiment 2. Antagonistic Effect of 2,4-D on Glyphosate Efficacy

Data were subjected to an ANOVA test using Genstat to test for treatment-by-experimental run interaction. Data were pooled across two experimental runs as differences between the two experimental runs were nonsignificant. All data met the assumption of homogeneity of variance. The significant difference in glyphosate dose response between glyphosate treatment alone and treatment with glyphosate plus 2,4-D for both populations was determined using LSD (*p* ≤ 0.05). Graphs were plotted in SigmaPlot 14.0.

## 3. Results and Discussion

### 3.1. Experiment 1. Effect of Temperature on Glyphosate Efficacy

Irrespective of temperature regimes, population Ch was found to be sensitive to the recommended dose of glyphosate (570 g ae ha^−1^; thereafter, g ha^−1^) compared with other populations. For the LT regime, there was no survival for SGM2 and SGW2 populations at 1140 g ha^−1^ glyphosate; however, at the HT regime, 28 and 37% of the plants survived the glyphosate application at this rate, respectively (Table 2; Figure 1). Similarly, for the CP2 population, 92 and 54% of plants survived the application of glyphosate at 1140 and 2280 g ha^−1^, respectively, when applied during the HT regime. Similar results were found for biomass production (Figure 1).

The biomass of Ch population was nil at the recommended glyphosate rate of 570 g ha^−1^ when plants were grown either at HT or LT regimes as plants were completely killed. However, the biomass of the SGM2 population at glyphosate 570 g ha^−1^ was reduced by 78 and 96% at HT and LT regimes, respectively, compared with their respective control (Figure 2). At the same glyphosate dose, biomass in SGW2 was reduced by 66 and 84% at HT and LT regimes, respectively, compared with their respective control (Figure 2). The CP2 population had poor control with glyphosate application even at 1140 g ha^−1^. Glyphosate 570 g ha^−1^ caused a reduction in CP2 biomass by only 44 and 78% at HT and LT regimes, respectively, compared with their respective control. LD_50_ and GR_50_ values for SGM2, SGW2, and CP2 were increased remarkably when plants were grown at HT (Table 2). LD_50_ values of glyphosate for SGM2, SGW2, and CP2 populations were 60, 654, and 1067 g ha^−1^, respectively, at LT, which increased to 557, 2108, and 2554 g ha^−1^, respectively, at HT regimes (Table 2 and Table 3). GR_50_ values of glyphosate for SGM2, SGW2, and CP2 populations at LT were 145, 252, and 259 g ha^−1^, respectively, which increased to 273, 446, and 733 g ha^−1^, respectively, at HT regimes (Table 3).

This study revealed that populations of *C. virgata* collected from Queensland differed in their resistance behavior to glyphosate application. On the basis of high LD_50_ and GR_50_ values and significant resistant factors, SGW2 and CP2 populations were found to be highly resistant to glyphosate compared with the GS Ch population. Further, populations showed a differential behavior to glyphosate between the LT and HT regimes. The GR_50_ value of the GS population Ch did not vary much at varied temperature regimes. However, in the case of GR populations, such as SGW2 and CP2, GR_50_ values of glyphosate at the HT regime increased by 1.8 to 2.8 times compared with the LT regime. Similarly, LD_50_ values of SGW2 and CP2 at the HT regime were increased by 3.2 and 2.4 times, respectively, compared with the LT regime. This suggests that glyphosate efficacy for the control of *C. virgata* was higher in the LT regime compared with HT, especially in GR populations. Greater control and biomass reductions in different weeds with glyphosate application were observed in earlier studies at LT regimes compared with HT regimes [14,16,25]. The physiological reason for the reduced control of GR populations, such as SGW2 and CP2, at the HT regime, is not known and needs to be determined. It is quite possible that populations of *C. virgata* may differ in growth rate for showing resistance behavior; however, we have not evaluated these facts in this study. Some authors postulated that the reduced control of GR weeds at HT might be due to the interaction of the resistant mechanism that leads to reduced sensitivity at high temperature [14,17]. For example, shikimate accumulation in leaf discs (marker of glyphosate inhibition of EPSPS) in some populations of *E. colona* was found different at 20 °C and 30 °C [14]. At high temperatures, a barrier may exist that could cause limited access of glyphosate to the chloroplast at HT compared with LT [14].

In a previous study, the efficacy of glyphosate against GR horseweed (*Conyza canadensis* (L.) Cronq.) was reduced at HT; whereas, no such phenomenon was observed in GS plants [26]. These authors postulated that GR plants grew more rapidly than GS plants, which could be the reason for the reduced efficacy of glyphosate on GR plants at HT. However, it is not known whether a similar phenomenon may occur in GR and GS *C. virgata* populations and what biochemical changes may occur in plants when both GR and GS plants are sprayed at a similar leaf stage. It would be interesting to investigate whether temperature plays a role in this particular target-site mutation. A recent study in Queensland revealed that glyphosate resistance in SGW2 and CP2 populations of *C. virgata* was due to a target-site-based resistance [12], and these authors did not rule out or look to confirm that other resistance mechanisms were present. Such a study would only be enlightening for investigating this effect of the presence of other mechanisms were ruled out. However, this increased resistance factor might be due to other mechanisms involved. It is also possible that higher temperatures could have promoted faster sequestration into a compartment or perhaps faster degradation of glyphosate (which has been seen in *E. colona* populations from Australia) was not investigated and this, cannot be ruled out. For example, in GR horseweed, it was found that cold temperatures slow down the sequestration of glyphosate in the vacuoles [17]. Contrary to this, in studies conducted in South Australia, it was found that the GR_50_ value of glyphosate for the control of junglerice increased at HT (30 °C) compared with LT (20 °C) [14]. These authors suggested that at HT, the absorption of the herbicide by leaves was reduced which could have reduced the ability of the herbicide to enter into the chloroplast. In another study, GR junglerice plants did not show glyphosate injury when exposed to HT (35/30 °C); however, plants died when subjected to LT (15/10 °C) [25].

Common lambsquarters (*Chenopodium album* L.) and horseweed were found to be less sensitive to glyphosate at temperature regimes of 32/26 °C compared with 18/12 °C [27]. These authors suggested that reduced translocation of glyphosate might be the reason for the low efficacy of glyphosate at HT. Similarly, it was reported that the resistance level in GR soybeans (*Glycine max* (L.) Merr.) was reduced at HT (35 °C) due to enhanced translocation of glyphosate to the meristematic regions compared with LT (15 or 25 °C) [28].

*Chloris virgata* is mainly a summer-season weed and it has a high seed production potential [9,29]. Growers in Australia prefer a no-till production system as it conserves soil and water. A no-till system favors the emergence of this weed under optimum soil moisture conditions. Growers in Queensland and New South Wales, in general, grow one crop in a calendar year and manage weeds during the fallow period. They rely on glyphosate for weed control during the fallow period and as preplant weed control. Our results suggest that during preplant and the summer fallow period (mean day temperature range 31–34 °C), growers may aggravate the problem of glyphosate failure. Poor control of *C. virgata* may increase the infestation of this weed, especially in resistant populations, and lead to high seed production. Therefore, for fallow conditions, it may be better to control *C. virgata* with glyphosate in the spring or autumn season compared with the summer season. Alternative herbicides such as glufosinate, haloxyfop, and clethodim, etc., can also be used for control of glyphosate-resistant *C. virgata* under fallow conditions [30].

This study was conducted under controlled environmental conditions. However, results may vary under field conditions, where light, humidity, and wind velocity may play a role in herbicide absorption and translocation. Therefore, further studies are required to evaluate the integrated effect of light, humidity, temperature, and water stress on glyphosate efficacy. Molecular studies based on gene expression under varying environmental factors may provide strong evidence to explain physiological mechanisms involved with the variable response of different populations to glyphosate rates.

### 3.2. Experiment 2. Antagonistic Effect of 2,4-D on Glyphosate Efficacy

The field-recommended 2,4-D rate of 700 g ae ha^−1^ was used in a tank mix with glyphosate. The GS population Ch was moderately controlled (67% mortality) by glyphosate at 741 g ha^−1^, whereas mortality was only 37% when glyphosate at this rate was tank-mixed with 2,4-D (Figure 3). Similarly, the mortality rate of the Ch population was 100% at glyphosate 1482 g ha^−1^; however, mortality was 81% when glyphosate at this rate was tank-mixed with 2,4-D. The antagonistic effect of 2,4-D on glyphosate efficacy was not observed when assessed for biomass reduction of the Ch population. This was mainly because plants were only grown for 28 d after spraying. For the Ch population, the biomass reduction was ~85% at a glyphosate rate of 370 g ha^−1^, and this reduction increased to 94 and 100% at glyphosate 741 g ha^−1^ and 1482 g ha^−1^, respectively (Figure 3).

For the GR population CP2, the survival rate was high (72%), even when glyphosate was applied at a very high rate of 3705 g ha^−1^ (Figure 4). The tank mix application of 2,4-D did not increase or decrease the efficacy of glyphosate for controlling *C. virgata* up to a rate of 2223 g ha^−1^. The mortality rate of the CP2 population was 28% at glyphosate 3705 g ha^−1^; however, not even a single plant was killed when glyphosate at this rate was tank-mixed with 2,4-D. The antagonistic effect of 2,4-D on glyphosate efficacy for controlling the CP2 population was more evident in the biomass reduction data (Figure 4). For the CP2 population, the biomass reduction was 90% at glyphosate 3705 g ha^−1^; however, this reduction was 64% when glyphosate at this rate was tank-mixed with 2,4-D (Figure 4).

Glyphosate provides a wide spectrum of weed control (grasses, sedges, and broadleaf weeds), whereas 2,4-D mainly controls dicot species. Farmers in Australia practice preplant or fallow weed control by tank mixing 2,4-D with glyphosate for effective control of grass as well as broadleaf weeds. In some weeds such as johnsongrass, wild oats (*Avena fatua* L.), and junglerice, tank-mix applications of 2,4-D with glyphosate resulted in reduced efficacy of glyphosate and provided poor weed control in previous studies [19,22,31]. In the current study, a 2,4-D and glyphosate antagonistic effect was confirmed for the GR population of *C. virgata*. An antagonistic effect was also found for the GS population (Ch) but at high glyphosate rates (741 and 1482 g ha^−1^). These results suggest that the 2,4-D-antagonism is dependent on the population’s sensitivity to glyphosate. There could be a mixed population of GR and GS *C. virgata* in paddocks, therefore, care should be taken with 2,4-D/glyphosate mixtures. In some studies, tank mixing 2,4-D (amine salt) with glyphosate caused reduced uptake and translocation of glyphosate in johnsongrass, kochia (*Kochia scoparia* L. (Schrad)), and waterhemp (*Amaranthus tuberculatus* (L.) Moq.) [22,31]. The mechanism for reduced uptake and translocation of glyphosate with auxin herbicides is not clear; however, it was suggested that it could be due to the chemical complexation through cation exchange or physical or physiological interactions [32,33].

In summary, the GR populations of *C. virgata*, such as SGW2 and CP2, were more sensitive to glyphosate at LT; therefore, the temperature should be considered during herbicide application, especially when applying preplant in the early spring or autumn season. However, the altered humidity level in the atmosphere may further influence results as this study was conducted at fixed humidity levels, which need further investigation. These results suggest that for better control of GR *C. virgata* populations, applications of glyphosate should be scheduled during cooler days (<25 °C) or during the evening to improve efficacy. In addition, for improved efficacy of glyphosate in *C. virgata,* temperature forecasts for the days following glyphosate application should be cooler. The prevalence of GR populations of *C. virgata* in glyphosate-tolerant crops such as cotton (*Gossypium hirsutum* L.) can be suppressed if glyphosate applications were made during cooler times. This study also highlighted that in a mixed population of weeds, and particularly when GR populations of *C. virgata* are prevalent in paddocks, it is advisable not to tank-mix glyphosate with 2,4-D. The results implied that the use of this mixture is useful if the feather fingergrass is not glyphosate-resistant; however, the use of the mixture is to be avoided if the population is glyphosate-resistant in order to not exacerbate the potential resistance problem.

A lack of full control of GR *C. virgata* in crops such as GR cotton may lead to poor competitiveness in the crop, reduced seed production, and, therefore, diminished weed seed banks in the soil. This study also highlights that there is a need to find a range of alternative herbicides for the control of GR populations of *C. virgata* in summer crops such as mungbean (*Vigna radiata* (L.) R. Wilczek), cotton, sorghum (*Sorghum bicolor* (L.) Moench), etc., and these herbicides may be integrated with agronomic tools (e.g., tillage regimes, narrow row spacing, high seeding rate, etc.) for sustainable weed control.

## 4. Conclusions

In summary, the GR populations of *C. virgata*, such as SGW2 and CP2, were more sensitive to glyphosate at LT; therefore, the temperature should be considered during herbicide application, especially when applying preplant in the early spring or autumn season. These results suggest that for better control of GR *C. virgata* populations, applications of glyphosate should be scheduled during cooler days (<25 °C) or during the evening to improve efficacy. This study suggests that a period of lower temperature forecast following glyphosate applications could improve efficacy, but this needs to be confirmed under field conditions. The prevalence of GR populations of *C. virgata* in glyphosate-tolerant crops such as cotton (*Gossypium hirsutum* L.) can be suppressed if glyphosate applications were made during cooler times. This study also highlighted that in a mixed population of weeds, and particularly when GR populations of *C. virgata* are prevalent in paddocks, it is advisable not to tank-mix glyphosate with 2,4-D. A lack of full control of GR *C. virgata* in crops such as GR cotton may lead to poor competitiveness in the crop, reduced seed production, and therefore diminish the weed seed banks in the soil. This study also highlights that there is a need to find a range of alternative herbicides for the control of GR populations of *C. virgata* in summer crops such as mungbean (*Vigna radiata* (L.) R. Wilczek), cotton, sorghum (*Sorghum bicolor* (L.) Moench), etc., and these herbicides may be integrated with agronomic tools (e.g., tillage regimes, narrow row spacing, high seeding rate, etc.) for sustainable weed control.

## Figures and Tables

**Figure 1 plants-11-03190-f001:**
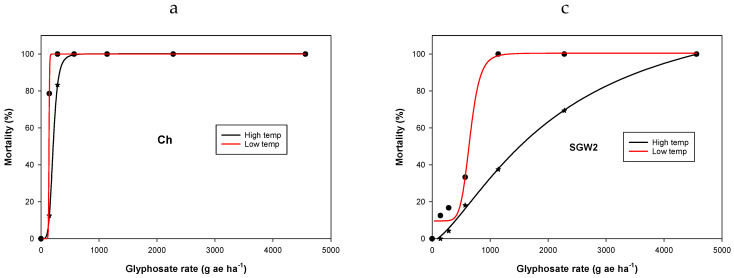
Dose–response curve of four populations ((**a**) Ch; (**b**) SGM2; (**c**) SGW2; (**d**) CP2) of *Chloris virgata* for plant mortality percentage. The curve is a log-logistic regression model fitted to data. Parameter estimates are provided in Table 2.

**Figure 2 plants-11-03190-f002:**
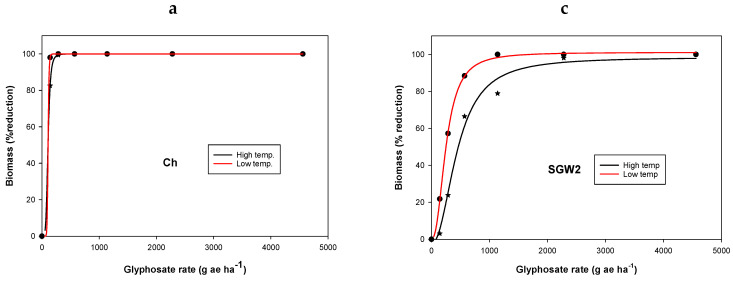
Dose–response curve of four populations ((**a**) Ch; (**b**) SGM2; (**c**) SGW2; (**d**) CP2) of *Chloris virgata* for percentage biomass reduction. The curve is a log-logistic regression model fitted to data. Parameter estimates are provided in Table 3.

**Figure 3 plants-11-03190-f003:**
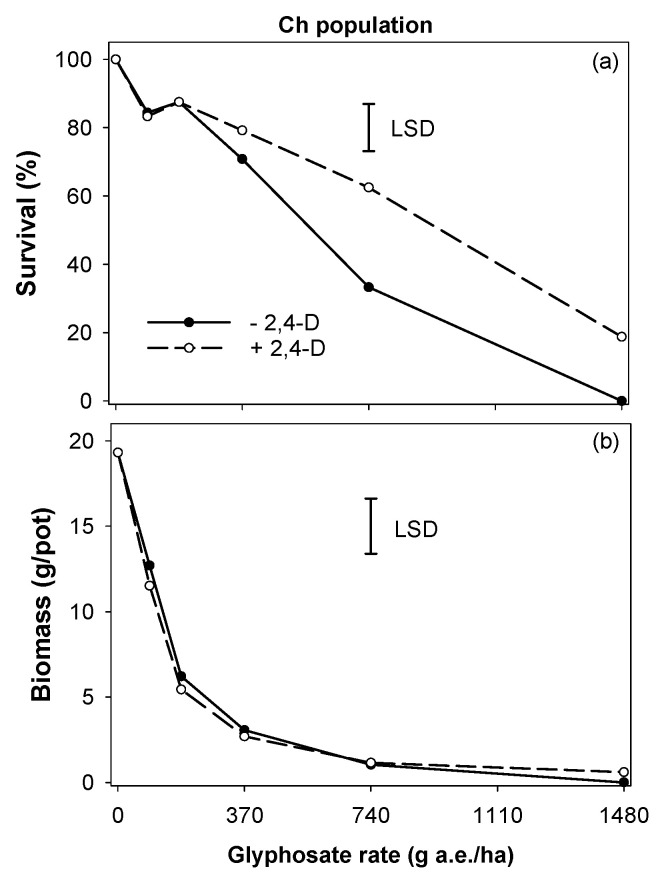
The antagonistic effect of 2,4-D on glyphosate efficacy for the Ch population of *Chloris virgata* in terms of (**a**) survival (%) and (**b**) biomass (g pot^−1^); LSD: least significant difference at the 5% level.

**Figure 4 plants-11-03190-f004:**
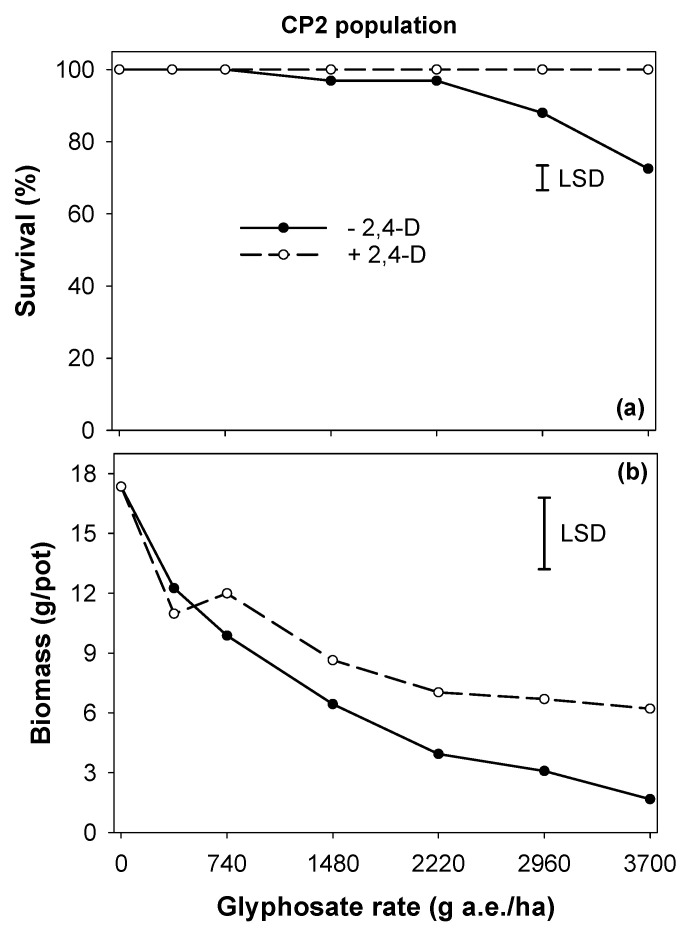
The antagonistic effect of 2,4-D on glyphosate efficacy for the CP2 population of *Chloris virgata* in terms of (**a**) survival (%) and (**b**) biomass (g pot^−1^); LSD: least significant difference at the 5% level.

**Table 1 plants-11-03190-t001:** Populations of *Chloris virgata* collected from different regions of Queensland, Australia.

Population	Location	Habitat	GPS Coordinates
Ch	Chinchilla	Wheat fallow	−26.8264, 150.5802
SGM2	St. George	Mungbean	−28.0916, 148.4271
SGW2	St. George	Wheat fallow	−28.0454, 148.3158
CP2	Cecil Plains	Sorghum	−27.2935, 151.1283

**Table 2 plants-11-03190-t002:** Estimates of regression parameters and doses required for 50% mortality (LD_50_) of *Chloris virgata* populations in dose–response studies conducted at high (35/25 °C) and low-temperature (25/15 °C) regimes in a glasshouse.

Population	Temperature Regimes	*y_0_*	*a*	*b*	LD_50_ g ha^−1^	Resistance Factor *	*R* ^2^
Ch	LT	−2.3 (2)	100 (3)	−29.2 (0.7)	137 (0.1)	-	0.99
	HT	0.01 (0.3)	100 (0.3)	−5.1 (0.1)	209 (0.8)	-	0.99
SGM2	LT	0.02 (6)	111 (23)	−0.6 (0.5)	60 (33)	-	0.98
	HT	−3.4 (6)	107 (12)	−1.6 (0.4)	557 (108)	2.7 a	0.99
SGW2	LT	9.6 (4)	91 (7)	−7.5 (9.3)	654 (117)	3.1 ab	0.99
	HT	−1.4 (1.4)	134 (12)	−1.4 (0.1)	2108 (275)	15.4	0.99
CP2	LT	1.5 (1.4)	100 (3)	−4.4 (0.6)	1067 (27)	5.1 ce	0.99
	HT	−0.2 (0.3)	119 (2)	−3.1 (0.1)	2554 (43)	12.2 d	0.99

Figures in parentheses indicate the standard error of means (±); LT: Low temperature; HT: High temperature; *y*_0_ = bottom of the curve; *a* = difference between the top and bottom of the curve; *b* = slope of the curve. * Values with different letters are significantly different at 5% level of significance on the basis of the *t*-test.

**Table 3 plants-11-03190-t003:** Estimates of regression parameters and doses required for 50% biomass reduction (GR_50_) of *Chloris virgata* populations in dose–response studies conducted at high (35/25 °C) and low-temperature (25/15 °C) regimes in the glass house.

Population	Temperature Regimes	*y_0_*	*a*	*b*	GR_50_ g ha^−1^	Resistance Factor *	*R* ^2^
Ch	LT	−4.8(0)	100 (0)	−13.2 (0.8)	107(2)	-	0.99
	HT	4.3 (0)	100 (0)	−5.1 (0.0)	106 (0.1)	-	0.00
SGM2	LT	−0.01 (0.5)	100 (0.6)	−2.6(0.1)	145 (1.4)	1.3 a	0.99
	HT	−3.9 (7)	101 (9.0)	−3(1.0)	273 (33)	2.6 b	0.97
SGW2	LT	0.3 (1.3)	101 (2.0)	−2.3 (0.1)	252 (6)	2.3 b	0.99
	HT	−2.3 (5.6)	101 (8)	−2.2(0.5)	446 (56)	4.2 c	0.98
CP2	LT	0.03 (2)	102 (3)	−1.7(0.1)	259 (13)	2.4 b	0.99
	HT	2.5 (4)	103 (9)	−1.6(0.3)	733 (98)	6.9 d	0.99

Figures in parentheses indicate the standard error of means (±); LT: low temperature; HT: high temperature; *y*_0_ = bottom of the curve; *a* = difference between the top and bottom of the curve; *b* = slope of the curve. * Values with different letters are significantly different at 5% level of significance on the basis of the *t*-test.

## Data Availability

All relevant data are within the manuscript.

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
