# Peer review of "Glyphosate Efficacy in Chloris virgata Sw. in Response to Temperature and Tank Mixing"

_plants, 2022, doi:10.3390/plants11233190_

Round 1
Reviewer 1 Report
This work focuses on experimental research of the environmental temperature or mixing glyphosate with 2,4-D effects on a problematic summer-season weed of Australia (Chloris virgata Sw.). The study raises an important issue in dealing with glyphosate-resistant weeds. The results obtained may be useful to many researchers interested in the antagonistic/synergistic effects of tank mixing on the suppression of weeds growth. The measurement technique is not fully described and should be supplemented with some details. The manuscript is moderately written, with minor typos. The manuscript can be published after some revision of the text.
Main remarks:
(l.46) To be able to compare with other weeds, an estimate of the area in million hectares infested by the glyphosate resistance C. virgata weed and/or a percentage relative to other weeds must be provided.
(l.97) Here is a typo - "contrllong".
(l.111) “…that were spread about 500 m in each paddock.” - Probably, 500 m^2 or 500x500 m^2 is meant here?
(l.131) It is necessary to clarify how the humidity changed depending on the temperature.
(l.129) “These treatments were tested in a factorial arrangement of a randomized complete block design in three replicates.”
(l.149) “All treatments were evaluated in a factorial arrangement of a randomized complete block design in three replicates by keeping four seedlings per pot.”
It is necessary to give a detailed description of what such testing means, to clarify the types of blocks and groups of conditions.
(l.156) It's not clear what this repetition means - “…respectively 30-33 ℃, respectively.”
(l.158) The 2,4-D form (amine/ether) or unmodified must be specified.
(l.204) For SGW2 37% is mortality.
(l.209) The parameter "Resistant factor" first appears in Table 2, it is necessary to define it earlier and describe how it was calculated.
(l.227-228) GR50 is not listed in Table 2.
(l.264) Do the considered populations of C. virgata differ in growth rate? I think readers might be interested too.
(l.284-287) Why is the term " Similarly" used here? Obviously, there are contradictions in these two sentences. "the low efficacy of glyphosate at HT" contradicts to "the resistance level in GR soybeans [Glycine max (L.) Merr.] was reduced at HT".
(l.353) Probably, the fact that 2,4-D is a salt of dimethylamine or an ether can play a significant role here.
(l.384-386) It should also be noted that in addition to C. virgata weed, there may be other harmful weeds in the paddock that are much more sensitive to tank mixes.
The experimental results presented in the manuscript show that increasing the concentration of glyphosate eventually has an inhibitory effect even on glyphosate-resistant weeds. In order to increase the reliability of the results and the validity of the conclusions, it is desirable in such experiments to monitor the change in the concentration of glyphosate in plant tissues and on the leaves, as well as the concentration of its degradation products. I wonder if there have been studies with similar monitoring?
Author Response
Response to Reviewer #1
Comments and Suggestions for Authors
This work focuses on experimental research of the environmental temperature or mixing glyphosate with 2,4-D effects on a problematic summer-season weed of Australia (Chloris virgata Sw.). The study raises an important issue in dealing with glyphosate-resistant weeds. The results obtained may be useful to many researchers interested in the antagonistic/synergistic effects of tank mixing on the suppression of weeds growth. The measurement technique is not fully described and should be supplemented with some details. The manuscript is moderately written, with minor typos. The manuscript can be published after some revision of the text.
Main remarks:
(l.46) To be able to compare with other weeds, an estimate of the area in million hectares infested by the glyphosate resistance C. virgata weed and/or a percentage relative to other weeds must be provided.
Response: The suggested information has been added.
(l.97) Here is a typo - "contrllong".
Response: Corrected.
(l.111) “…that were spread about 500 m in each paddock.” - Probably, 500 m^2 or 500x500 m^2 is meant here?
Response: Corrected.
(l.131) It is necessary to clarify how the humidity changed depending on the temperature.
Response: The suggested information has been added.
(l.129) “These treatments were tested in a factorial arrangement of a randomized complete block design in three replicates.”
(l.149) “All treatments were evaluated in a factorial arrangement of a randomized complete block design in three replicates by keeping four seedlings per pot.”
It is necessary to give a detailed description of what such testing means, to clarify the types of blocks and groups of conditions.
Response: The suggested information has been added.
(l.156) It's not clear what this repetition means - “…respectively 30-33 ℃, respectively.”
Response: Corrected.
(l.158) The 2,4-D form (amine/ether) or unmodified must be specified.
Response: It was amine salt. The information has been added.
(l.204) For SGW2 37% is mortality.
Response: No it was survival percentage. It means If mortality is 37% ; then the survival percentage is 63%
(l.209) The parameter "Resistant factor" first appears in Table 2, it is necessary to define it earlier and describe how it was calculated.
Response: The information has been added.
(l.227-228) GR50 is not listed in Table 2.
Response :GR50 values were mentioned in Table 3.
(l.264) Do the considered populations of C. virgata differ in growth rate? I think readers might be interested too.
Response: The suggested information has been added.
(l.284-287) Why is the term " Similarly" used here? Obviously, there are contradictions in these two sentences. "the low efficacy of glyphosate at HT" contradicts to "the resistance level in GR soybeans [Glycine max (L.) Merr.] was reduced at HT".
Response: Corrected.
(l.353) Probably, the fact that 2,4-D is a salt of dimethylamine or an ether can play a significant role here.
Response: It was amine salt and the information has been added,
(l.384-386) It should also be noted that in addition to C. virgata weed, there may be other harmful weeds in the paddock that are much more sensitive to tank mixes.
The experimental results presented in the manuscript show that increasing the concentration of glyphosate eventually has an inhibitory effect even on glyphosate-resistant weeds. In order to increase the reliability of the results and the validity of the conclusions, it is desirable in such experiments to monitor the change in the concentration of glyphosate in plant tissues and on the leaves, as well as the concentration of its degradation products. I wonder if there have been studies with similar monitoring?
Response: The suggested information has been added.
Response to Reviewer #2
Comments and Suggestions for Authors
Dear author(s),
there are some inspiring insights thorough the manuscript and I tend to agree on its publication. However, there are few points that needs to be quickly addressed to improve its overall communication:
Response: Please recheck the comments. I think these comments are related with other manuscript and attached here by mistake. For example in the Results comments : It was written : Does the life cycle affect earnings management and bankruptcy?" and "Quo Vadis, earnings management? Analysis of manipulation determinants in Central European environment"
Therefore, these are not related with this manuscript.
Title:
1/ clearly condensate the novelty and significance of the main discovery into a short and groundbreaking claim
Abstract:
2/ better follow the established schema of writing academic Abstract: A/ introduction (urgency and significance of the research hypothesis); B/ principles of the methods used + key results; C/ conclusions (commercial and environmental impacts)
3/ reduce the use of abbreviations and technical terms, please understand that the purpose of the Abstract is to explain to all readers (including those from other disciplines) what the paper is about
4/ there is no reason to go into detail and present the results obtained under specific reaction conditions, rather provide a synthesis of the results obtained
5/ better highlight the urgency and significance of your work, clearly indicate who (and how) will benefit from these findings (explain the importance)
introduction:
6/ significantly reduce sentences whose content you do not consider important enough to support with appropriate citations, remove all clusters of references to avoid reference overkill (prefer only 1 reference to support 1 claim)
7/ deeper review the latest trends in plant production, refer to papers "Silica nanoparticles from coir pith synthesized by acidic sol-gel method improve germination economics" and "Recovering phosphorous from biogas fermentation residues indicates promising economic results"
8/ go straight to the point and more in depth, write more technically (always provide corresponding numbers), significantly condensate all the text by reducing ballast phrases and cliché
9/ it should be highlighted that improving the condition of plants increases their resistance to disease vectors and parasites, refer to papers "Techno-economic analysis reveals the untapped potential of wood biochar" and "Economic considerations on nutrient utilization in wastewater management"
10/ make sure that this chapter fully introduces any reader into to the topic, explain all the terms, units, abbreviations and the whole context that is necessary for anyone (including experts from other disciplines) to understand the following chapters
Materials and Methods:
11/ the method must be presented in such a way that it can be reproduced anytime, by anyone, anywhere (do not create obstacles like referring to specific location etc.)
12/ each material/reactant and apparatus used needs to be presented in detail (serial number, setup, process parameters, manufacturer, country of origin, purity etc.)
13/ provide cost breakdown or at least some simplified financial analysis if you are about to argue that this concept is realistic
Results and Discussion:
14/ each Tab. and Fig. should be provided with caption that describes A/ what can be seen and B/ how is this relevant to the research hypothesis
15/ do not ignore (economic) reality, refer to papers "Does the life cycle affect earnings management and bankruptcy?" and "Quo Vadis, earnings management? Analysis of manipulation determinants in Central European environment"
16/ show more self-criticism to your work (can all the methods and results be fully trusted? what are the weaknesses of the methods used? where do the main measurement inaccuracies arise? what are the limitations from a commercial point of view? are the lessons learned transferable to other fields?)
17/ avoid data overkill, present only the most most industrially important results (refer to papers "Data-driven Machine Learning and Neural Network Algorithms in the Retailing Environment: Consumer Engagement, Experience, and Purchase Behaviors" and "Sustainable Organizational Performance, Cyber-Physical Production Networks, and Deep Learning-assisted Smart Process Planning in Industry 4.0-based Manufacturing Systems")
18/ compare your results in more depth with the existing literature, identify the main deviations and try to explain the mechanisms by which they may have been caused
19/ propose some improvements and direction for future research, get inspired in papers "Novel sorbent shows promising financial results on P recovery from sludge water" and "Two‐fraction anaerobic fermentation of grass waste"
18/ reveal the main driving mechanisms of your results, provide deeper synthesis and reveal some more original/significant findings
Conclusions:
19/ do not repeat your methods and results again and again, please understand that the Conclusion chapter is not a summary of your work, present only original and industrially significant revelations that have the potential to expand the horizon of human knowledge (higher level of generalization is mandatory)
20/ clearly indicate whether the research hypotheses tends to be confirmed or not
Reviewer 2 Report
Dear author(s),
there are some inspiring insights thorough the manuscript and I tend to agree on its publication. However, there are few points that needs to be quickly addressed to improve its overall communication:
Title:
1/ clearly condensate the novelty and significance of the main discovery into a short and groundbreaking claim
Abstract:
2/ better follow the established schema of writing academic Abstract: A/ introduction (urgency and significance of the research hypothesis); B/ principles of the methods used + key results; C/ conclusions (commercial and environmental impacts)
3/ reduce the use of abbreviations and technical terms, please understand that the purpose of the Abstract is to explain to all readers (including those from other disciplines) what the paper is about
4/ there is no reason to go into detail and present the results obtained under specific reaction conditions, rather provide a synthesis of the results obtained
5/ better highlight the urgency and significance of your work, clearly indicate who (and how) will benefit from these findings (explain the importance)
introduction:
6/ significantly reduce sentences whose content you do not consider important enough to support with appropriate citations, remove all clusters of references to avoid reference overkill (prefer only 1 reference to support 1 claim)
7/ deeper review the latest trends in plant production, refer to papers "Silica nanoparticles from coir pith synthesized by acidic sol-gel method improve germination economics" and "Recovering phosphorous from biogas fermentation residues indicates promising economic results"
8/ go straight to the point and more in depth, write more technically (always provide corresponding numbers), significantly condensate all the text by reducing ballast phrases and cliché
9/ it should be highlighted that improving the condition of plants increases their resistance to disease vectors and parasites, refer to papers "Techno-economic analysis reveals the untapped potential of wood biochar" and "Economic considerations on nutrient utilization in wastewater management"
10/ make sure that this chapter fully introduces any reader into to the topic, explain all the terms, units, abbreviations and the whole context that is necessary for anyone (including experts from other disciplines) to understand the following chapters
Materials and Methods:
11/ the method must be presented in such a way that it can be reproduced anytime, by anyone, anywhere (do not create obstacles like referring to specific location etc.)
12/ each material/reactant and apparatus used needs to be presented in detail (serial number, setup, process parameters, manufacturer, country of origin, purity etc.)
13/ provide cost breakdown or at least some simplified financial analysis if you are about to argue that this concept is realistic
Results and Discussion:
14/ each Tab. and Fig. should be provided with caption that describes A/ what can be seen and B/ how is this relevant to the research hypothesis
15/ do not ignore (economic) reality, refer to papers "Does the life cycle affect earnings management and bankruptcy?" and "Quo Vadis, earnings management? Analysis of manipulation determinants in Central European environment"
16/ show more self-criticism to your work (can all the methods and results be fully trusted? what are the weaknesses of the methods used? where do the main measurement inaccuracies arise? what are the limitations from a commercial point of view? are the lessons learned transferable to other fields?)
17/ avoid data overkill, present only the most most industrially important results (refer to papers "Data-driven Machine Learning and Neural Network Algorithms in the Retailing Environment: Consumer Engagement, Experience, and Purchase Behaviors" and "Sustainable Organizational Performance, Cyber-Physical Production Networks, and Deep Learning-assisted Smart Process Planning in Industry 4.0-based Manufacturing Systems")
18/ compare your results in more depth with the existing literature, identify the main deviations and try to explain the mechanisms by which they may have been caused
19/ propose some improvements and direction for future research, get inspired in papers "Novel sorbent shows promising financial results on P recovery from sludge water" and "Two‐fraction anaerobic fermentation of grass waste"
18/ reveal the main driving mechanisms of your results, provide deeper synthesis and reveal some more original/significant findings
Conclusions:
19/ do not repeat your methods and results again and again, please understand that the Conclusion chapter is not a summary of your work, present only original and industrially significant revelations that have the potential to expand the horizon of human knowledge (higher level of generalization is mandatory)
20/ clearly indicate whether the research hypotheses tends to be confirmed or not
Author Response

(The authors gave the same response as above.)
